# Neural Activities Classification of Human Inhibitory Control Using Hierarchical Model

**DOI:** 10.3390/s19173791

**Published:** 2019-09-01

**Authors:** Rupesh Kumar Chikara, Li-Wei Ko

**Affiliations:** 1Department of Biological Science and Technology, College of Biological Science and Technology, National Chiao Tung University, Hsinchu 300, Taiwan; 2Center for Intelligent Drug Systems and Smart Bio-devices (IDS2B), National Chiao Tung University, Hsinchu 300, Taiwan; 3Institute of Bioinformatics and Systems Biology, National Chiao Tung University, Hsinchu 300, Taiwan; 4Swartz Center for Computational Neuroscience, University of California San Diego, San Diego, CA 92093, USA

**Keywords:** electroencephalography (EEG), ERP-P300, hierarchical classification model, phase locking value, brain-computer interface, human inhibitory control

## Abstract

Human inhibitory control refers to the suppression of behavioral response in real environments, such as when driving a car or riding a motorcycle, playing a game and operating a machine. The P300 wave is a neural marker of human inhibitory control, and it can be used to recognize the symptoms of attention deficit hyperactivity disorder (ADHD) in human. In addition, the P300 neural marker can be considered as a stop command in the brain-computer interface (BCI) technologies. Therefore, the present study of electroencephalography (EEG) recognizes the mindset of human inhibition by observing the brain dynamics, like P300 wave in the frontal lobe, supplementary motor area, and in the right temporoparietal junction of the brain, all of them have been associated with response inhibition. Our work developed a hierarchical classification model to identify the neural activities of human inhibition. To accomplish this goal phase-locking value (PLV) method was used to select coupled brain regions related to inhibition because this method has demonstrated the best performance of the classification system. The PLVs were used with pattern recognition algorithms to classify a successful-stop versus a failed-stop in left-and right-hand inhibitions. The results demonstrate that quadratic discriminant analysis (QDA) yielded an average classification accuracy of 94.44%. These findings implicate the neural activities of human inhibition can be utilized as a stop command in BCI technologies, as well as to identify the symptoms of ADHD patients in clinical research.

## 1. Introduction

The ability to stop ongoing motor action has been recognized as an inhibition of human response or inhibitory control. The ability to inhibit unsuitable action is a symbol of human motor control. The human inhibition process was usually studied in the laboratory setting by using a Go/No-Go or stop- signal task. The stop-signal task has been famed for investigating the neural marker of human inhibition [1]. The EEG-P300 neural marker of human inhibition are attracting increased attention owing to their reputed importance in rehabilitation and clinical research, such as attention deficit/hyperactivity disorder [2,3]. The EEG-P300 wave has been frequently used as a neural marker to examine the inhibitory control of both healthy people and group of ADHD patients. The preceding EEG study reported that EEG-P300 signals were commonly used as a neural marker for response inhibition [4,5]. Several EEG-based studies have addressed the relationship between P300 and neural oscillations of human inhibitory control [4,5,6]. Some studies reported that neural oscillations of inhibition induced event related potentials (ERP) P300 waves in the frontal lobe [5,6,7]. These neural markers in a stop signal task have always been of interest because they can be used to recognize the human inhibitory control. Event-related potential (ERP) was evoked brain response that synchronized by target stimulus. The ERP-P300 wave was a positive potential, it was induced 300 ms after a stop stimulus [7,8]. The P300 wave, first described by Sutton et al. [7] during human inhibition and selective attention. In addition, former study reported that the P300 wave after the onset of a stop stimulus in a stop-signal task has been linked to the human inhibition in the frontal-central brain regions [9]. One more study analyzed the effect of human inhibitory control by determining P300 responses in stop-signal task [10]. An earlier study used F3, FZ, F4 electrodes on frontal lobe and C3, CZ, C4 electrodes on motor cortex they found N200 and P300 waves in human inhibition [11,12]. However, our study suggest that ERP-P300 wave classification can be used as a clinical evaluation of patients with ADHD. 

In addition, another importance of this study is in the development of BCI systems. BCI technology is a fast growing area of research. It has important contributions in the medical field, such as the rehabilitation for paralyzed people. Mind reading and remote communication have their exclusive impression in various research fields, like wheelchair, robotic arm and drone control. In current BCI technology, devices such as a wheelchair, robotic arm and drone can easily turn left, turn right, move forward based on left-hand and right-hand motor imagery (MI) [4,6], but these mental states cannot stop the movement of BCI devices. What kind of brain signals can directly stop robotic arm, drone and wheelchair movement? To address this problem, the mental state of human inhibition can be used as a stop command. The neural activities related to inhibition (P300) can be applied as a stop command in current BCI technology. Moreover, the BCI research area has grown swiftly in recent years, allowing the development of faster and more reliable assistive tools based on direct relations between the human brain and machine. The BCI system depends on the recognition and classification of event related potentials that were generated by the human brain response. Additionally, in BCI technologies, changes in human brain activities were identified using various classification algorithms, including the popular machine learning method of linear discriminant analysis [13,14,15]. In this work, we used high temporal resolution EEG method because this method has the benefit of being non-invasive, low-cost and moveable. It is a very popular technique to use between the BCI area and clinical research [4,5]. An EEG was generated by extracellular current that was related to synaptic potentials of the pyramidal cells of the cerebral lobe. EEG has excellent temporal resolution [6,16,17]. Finally, EEG is a noninvasive and portable technique for recording brain activities at various locations on the scalp map [18,19,20]. In this study, EEG signals were obtained using thirty-two EEG electrodes. All EEG electrode locations were organized according to the international 10–20 system.

In this work, for the first time we used brain connectivity by phase locking value (PLV) method in eight pairs of electrodes included F3-F4, F4-O1, F4-T8, T7-O1, T8-O2, T7-T8, C3-CZ, C4-CZ, which covered most of the human brain areas related to inhibition, as well as frontal lobe, left motor cortex, right motor cortex, left temporal lobe, right temporal lobe and occipital lobe. We selected these eight brain regions of interest to identify the neural markers of inhibition. We used PLV because this method has revealed the best performance of the classification system. The PLV method has been used as an important feature for the classification of EEG signals [21]. Our study used the brain connectivity method to hierarchically classify the neural activities of left hand response (LHR) and right hand response (RHR) inhibitions. We hypothesize that this hierarchical model with brain connectivity method will achieve higher accuracy in the brain cortex related to human inhibition, such as the frontal and temporal-occipital lobes. The main impact of this work is the development of a hierarchical classification model to recognize the neural activities of human inhibition in successful-stop (SS) vs. failed-stop (FS) trials. These neural activities can be used as a stop command in the BCI technologies, such as in the stopping of the movement of robotic arm, drone and wheelchair. Also, these neural marker can be applied to recognize the symptoms of ADHD patients in clinical research. 

## 2. Materials and Methods

### 2.1. Subjects

Twelve healthy male subjects participated in the stop-signal task or Go/No-Go task. All subjects used response inhibitions of the right-hand and left-hand. All subjects were between 25–30 years old and had normal vision. None had experience in the stop-signal task before participating in left- and right-hand response inhibitions. No subject had a history of psychological disorders. Before the experiment, each participant provided a written informed consent. This study was approved by the Research Ethics Committee of the National Taiwan University, Taipei, Taiwan. The experimental protocol was carried out in accordance with the laws of the country and was approved by the institutional review board (IRB) of the National Taiwan University, Taipei, Taiwan.

### 2.2. Experimental Scenario

In this study, all subjects performed a stop-signal task or Go/No-Go task [1]. The stop signal task is widely used to measure the inhibition of the human response and the main task dependent variable, the stop signal reaction time (SSRT), provides an individualized measure of inhibitory control. The participants were presented with a series of Go stimuli they were instructed to respond quickly; for a circle or square shape. They were instructed to press the right button for a square and a left button for a circle, as presented in Figure 1. 

In a subset of trials, the Go stimulus was followed, after a variable delay, by a stop signal (i.e., a beep), to which participants were instructed to inhibit their response. In 75% go trials, merely a square and a circle stimuli were presented, and subjects were instructed to answer to the stimulus rapidly and accurately as possible. In 25% stop trials, the square and the circle stimuli were followed by an auditory stop signal (beep tone, 100 ms), and subjects were instructed to withhold their responses in auditory stop trial. The fixation sign and a square and a circle stimuli were displayed in the center of the computer screen, in white, on a black background, as shown in Figure 1. Each go trial and stop trial was started with the display of a white central fixation cross, which was followed by a square and a circle shape (see Figure 1). In a randomly selected 25% of the stop trials, subjects received an auditory tone binaurally by headphones. This short beep tone lasting for 100 ms, it was used as a stop-signal and prompting subjects to inhibit their left-hand response or right-hand response in the stop trial. Stop signals were presented the same number of times following each symbol. In stop trials, a stop signal was presented after a variable stop-signal delay (SSD), which was initially set to 250 ms, and it was adjusted using the staircase tracking procedure. The delay between the appearance of the go stimulus (i.e. visual stimulus) and the auditory stop signal varied trials with trials using a staircase method, which either increased or reduced the SSD by 50 ms in the next stop trial, depending on whether the subject had successfully inhibited (SSD increased) or failed to inhibit (SSD decreased). This one-up and one-down tracking procedure ensures that participants inhibit on approximately half of all trials and controls for difficulty level across participants. In this study, each subject performed 100 trails.

### 2.3. Acquisition of EEG Signals

The EEG signals were acquired using a 32–channel EEG cap with Ag/AgCl electrodes, which were located in 10–20 system. We used a Scan NuAmps Express system (Compumedics USA Inc., Charlotte, NC, USA). The sampling rate was 1000 Hz. To reduce the number of data and eliminate noise, the data was down-sampled from 1000 Hz to 500 Hz and filtered using a 1–50 Hz bandpass IIR filter before further analysis. Independent component analysis (ICA) has been an effective method to eliminate various artifacts, such as eye movement, eye blink artifacts and muscle artifacts [22]. Therefore, the ICA method was used to extract pure EEG signals from the human brain. In this study, we used the runica algorithms of Infomax ICA decomposition in MATLAB R2012b and EEGLAB toolbox [23]. To identify various types of artifacts, we look at the scalp map, the power spectrum and the location of the dipole sources of each independent component. Based on these criteria, we separate good and bad independent components, back-projecting the retained components to clean the EEG signal. Subsequently, we used clean EEG signals to measure ERP-P300 wave and phase-locking value (PLV) during LHR and RHR inhibitions. The acquired clean EEG signals were extracted in successful stop (SS) and failed stop (FS) stop trials (from −200 ms to 1300 ms). In addition, the EEG signals were segmented after the onset of the stimulus from 1 ms–300 ms to obtain the phase-locking value (PLV). The extracted EEG signals were used to classify ERP-P300 waves using PLV as a feature during the SS and FS trials in inhibitions of LHR and RHR. The ERP-P300 wave has been recognized as a neural marker of human response inhibition. Figure 2 presents all of the steps of the EEG signal analysis. In this study, the statistical significance (*p* < 0.05) difference in the P300 wave between the successful stop and failed stop trials was investigated with the Wilcoxon signed-rank test. In addition, the one-way ANOVA was used to match the significant difference (*p* < 0.01) in the classification accuracy between four groups QDA, KNNC, PARZENDC and LDA during LHR and RHR inhibitions.

### 2.4. Phase Locking Value (PLV)

Figure 3 shows the PLV method was used to quantify the neural activities between the EEG signals of two electrodes. Previous studies reported that PLV has been used to measure the brain connectivity between two EEG signals [24,25]. Another previous study showed that PLV is recognized as the best feature for the classification of EEG signals, and shows the good performance of the classification system [21]. The PLV method has been used to examine the synchronization and desynchronization between two EEG signals and revealed the phase-amplitude coupling between two EEG signals [21,25]. Therefore, in our study, we used PLV as an optimal feature to classify the successful stop (SS) and failed stop (FS) trials during RHR and LHR inhibitions. Equation (1) presents the PLV process between two EEG signals:(1)$$\mathrm{PLV}=\frac{1}{N}\overset{N}{\underset{k=1}{\sum }}e^{j}\left|\varphi 1\left(k\right)-\varphi 2\left(k\right)\right|$$
where *N* is the number of samples in the measured time window. The *φ*1 and *φ*2 are the phase values of the two EEG signals. The always normalized PLV has fluctuated between 0 and 1. A PLV value of 0 means that the phase of the two EEG signals is not synchronized, although a PLV of value 1 means that the two EEG signals are perfectly synchronized. These phase-locking values were used as features for classifiers.

The complete analysis of PLV is described in Figure 3, as follows: (I) Filtering: the EEG signal was first filtered in 1–50 Hz frequency by an infinite impulse response (IIR) filter before further analysis. The sampling rate of the EEG signal was used 500 Hz for PLV. (II) Hilbert transform: this step was used to quantify rising and falling of EEG signals. Hilbert transform of a signal was used to calculate the instantaneous amplitude as well as the instantaneous phase [24]. We used phase (*φ*) for PLV computation. Phase (*φ*) was a value between −*π* and *π*. (III) PLV metric: after completing the Hilbert transformation analysis, we observed a phase time course *φ* for each EEG signal. After that, we calculated the phase time courses of two EEG signals. The difference between these two-time courses (Δ*φ*) quantifies locking between the phases of these two signals. See Equation (1) in the analysis scheme to understand this process [25]. (IV) Normalization: for the normalization process, we used the pre-stimulus time, as the baseline and perform a z-transform normalization method to normalize the EEG signals. In this work, PLVs were utilized as the optimal features with parametric [linear discriminant analysis (LDA), quadratic discriminant analysis (QDA)] and nonparametric [parzen density-based classification (PARZENDC), K-nearest neighbor classification (KNNC)] classification algorithms to classify EEG data from successful stop (SS) and failed stop (FS) trials during RHR and LHR inhibitions. However, without the coupled brain regions, high-dimension feature vectors would be obtained and a complicated analysis would be made [21]. Therefore, in this study, the PLV method was used for selected coupled brain regions. This method has the benefits of good preparation of EEG channels and fast processing of EEG data in the hierarchical classification model.

### 2.5. Feature Extraction for Hierarchical Classification Model

As part of the feature extraction process, the PLV was measured between eight pairs of EEG-channels, which were included F3-F4, F4-O1, F4-T8, T7-O1, T8-O2, T7-T8, C3-CZ, and C4-CZ. The PLV output in successful stop (SS) and failed stop (FS) trials was used as input features in the parametric (LDA, QDA) and nonparametric (PARZENDC, KNNC) classification algorithms. Most previous studies used EEG spectral power as a feature to classify EEG signals during the stop signal task [10,25,26,27]. Therefore, in this study, we propose for the first time a hierarchical classification model with PLV (i.e., coupling method) as input features. In proposed hierarchical classification model we used parametric (LDA, QDA) and nonparametric (PARZENDC, KNNC) classifier algorithms to classify the EEG signals from successful stop (SS) and failed stop (FS) trials during both RHR and LHR inhibitions.

### 2.6. Forward Feature Selection for Hierarchical Classification Model

The feature selection process is called variable selection or attribute selection. Feature selection refers to the selection of a subset of related features for used in creating a model. Feature selection systems were used for three main reasons, which were the generalization of models to make them easier to understand, the shortening of model training times, and the improvement of generalization by reducing overfitting [28]. The data set contains many features that were either redundant or inappropriate and so be removed without significant loss of information [2]. Redundant features differed from unrelated features because a related feature may be redundant in the presence of an additional related feature with which it was strongly correlated [29]. In this study, we used forward feature selection method started with a blank set, X = 0, to which the most significant features with respect to X were added. Forward feature selection was used to select the best PLV features during human response inhibition. Moreover, forward feature selection procedure began by evaluating all subsets of features that consist of a single input attribute. In other words, we started by measuring the leave-one-out cross validation (LOOCV) error of the one component subsets, {X1}, {X2}, {XM}, where M is the input dimensionality. Consequently, we found the best individual feature, X(1).

### 2.7. Parametric and Nonparametric Classifier Algorithms in Hierarchical Model

In this study, four different, parametric (LDA, QDA) and non-parametric (PARZENDC, KNNC) classifiers were adopted to classify EEG data (i.e., P300) from successful stop (SS) and failed stop (FS) trials during RHR and LHR inhibitions. Using these classifiers, we observed which classification had optimal performance for inhibition-based P300 classification. The linear discriminant analysis (LDA) was originally developed in 1936 by Fisher [30]. LDA is a popular method of linear classification widely applied in statistics and machine learning. The average vectors and covariance matrices of different classes are the main parameters of LDA, which are calculated to find the appropriate discrete features and separate them into two or more classes. Moreover, a quadratic discriminant analysis (QDA) classifier has been used in machine learning and statistical classification to separate measurements of two or more classes of objects or events by a quadric surface. It is a more general version of the LDA classifier. QDA is a common multivariate classification similar to LDA. Unlike LDA, which uses a linear boundary between data points of different classes, QDA separates the estimates of two or more classes with a quadratic surface [31]. The k-nearest-neighbor classifier (KNNC) is one of the most basic classifiers for P300 signals. An unlabeled testing data point is classified by estimating the k neighbors (k = 10 in this work) nearest to the testing data point among the training samples. In addition, PARZENDC has been used as a nonparametric density estimation method. It was used for EEG-P300 signals classification. Using a given kernel function, the technique approximates a given training set distribution using a linear combination of kernels that are centered on the observed points. In this work, densities of each two classes were separately approximated and a test point was assigned to the class with maximal posterior probability [32]. The Parzendc design procedure is based on a Parzen kernel density estimate (Parzen, 1962 [33] and Webb, 2002 [34], with the value of its smoothing kernel parameter got as a maximum probability estimation. The hierarchical classification accuracies of the inhibition-based P300 signals were calculated as the mean of the classification result for each sample. Statistical significance was investigated by one-way ANOVA test, with significance indicated by a *p*-value lower than 0.01 (*p* < 0.01).

### 2.8. Accuracy Estimation Method 

Cross validation is a statistical technique to evaluate and compare learning algorithms by splitting the data into two parts: one used to train a model and the other used to validate the model. In classic cross-validation, training and validation sets must be crossed in successive rounds such that each data point has the possibility of being validated. The accuracy of the proposed model was evaluated by leave-one-out cross-validation method [35]. This method reduces the probability of obtaining erroneous results, as it studies multiple splits of the EEG data. In this study, the EEG data was assigned 90% of trials for training and 10% of trails for testing to investigate the performance of the proposed hierarchical classification model.

## 3. Results

In this study, a hierarchical classification model with a brain connectivity method was proposed to classify the EEG signals of the successful stop (SS) and failed stop (FS) trials during RHR and LHR inhibitions in the across subjects. In a hierarchical classification model, the parametric and non-parametric classification algorithms were used to classify the EEG-P300 signals of SS and FS trials under inhibitions of both LHR and RHR. General parametric machine learning algorithms were used, including linear discriminant analysis (LDA) classifier and quadratic discriminant analysis (QDA) classifier. The common nonparametric machine learning algorithms were used, including Parzen density-based classifier (PARZENDC) and K-nearest neighbor classifier (KNNC). In this investigation, basic parametric and nonparametric classification algorithms were used to evaluate hierarchical system performance, because these are very easy to use. If these basic classifiers perform batter with high accuracy, then the use of more complex classification algorithms could be avoided.

### 3.1. ERP-P300 Wave during Human Inhibitory Control

Figure 4A shows the grand average event-related potential (ERP) image of P300 waves arisen as a result of neural activities, which were involved in RHR and LHR response inhibitions. Accordingly, the brain dynamics of the P300 wave were observed in the frontal lobe (Fz) of the brain during successful stop (SS) and failed stop (FS) trials in both LHR and RHR inhibitions. The ERP-P300 wave (i.e., positive potential) appeared around 300 ms after RHR and LHR inhibitions in all subjects, as shown in Figure 4. Moreover, Figure 4B displays the neural activities of LHR inhibition during successful stop (SS) and failed stop (FS) trials. We observed the ERP-P300 wave in the frontal lobe (Fz) of the brain during LHR inhibition. The ERP-P300 wave is a neural signature of human response inhibition. These neural activities revealed that all subjects successfully followed the stop-signal task. Acquired ERP-P300 waves shown higher positive potentials during RHR and LHR inhibitions in the failed stop (FS) trials than in the successful stop (SS) trails. Figure 4C shows the higher positive potential of the P300 wave in RHR (SS) than in LHR (SS) that were observed during the successful stop (SS) trials. 

The statistical significant (*p* < 0.05) difference in ERP-P300 waves between successful stop (SS) and failed stop (FS) trials were investigated using Wilcoxon signed rank test. The green asterisk display significant (*p* < 0.05) difference between successful stop (SS) and failed stop (FS) trials, as shown in Figure 4A–C. In this study, the ERP-P300 waves were used as the neural signature of human response inhibition of left-hand or right-hand. The current EEG study clearly presented the response inhibition-related P300 waves in the frontal lobe of the brain during RHR and LHR inhibitions. The ERP-P300 waves during successful stop (SS) and failed stop (FS) trials were used to calculate the PLV between two EEG electrodes signals. Finally, the PLV was used as an input feature in the hierarchical classification modal. The PLV results are described in the next section.

### 3.2. EEG-PLV in Human Inhibitory Control

PLVs were calculated between the EEG signals of two electrodes in two different brain regions to classify the neural activities of RHR and LHR inhibitions. The ERP-P300 signals that all participants demonstrated in the left and right response inhibitions were used for PLV analysis. The PLV vector defined in the previous section was used as the input feature for both parametric (LDA, QDA) and nonparametric (PARZENDC, KNNC) classification algorithms. The PLV method was used to select coupled brain regions related to inhibition of the human response because the PLV method has shown the best performance of the classification system [25]. PLVs were computed for eight pairs of EEG channels including F3-F4, F4-O1, F4-T8, T7-O1, T8-O2, T7-T8, C3-CZ, and C4-CZ. These eight pairs of EEG channels were used as a region of interest (ROI) because their function role in response inhibition, such as the frontal lobe (F3, F4) has been associated to human inhibition; the motor cortex (C3, Cz, C4) has been associated with motor functions (left-hand or right-hand movements); the temporal lobe (T7, T8) has been related to auditory stimulation, and the occipital lobe (O1, O2) has been related to visual stimuli. In this study, the leave-one-out method was used to compare the accuracies of the parametric (LDA, QDA) and nonparametric (PARZENDC, KNNC) classification algorithms [35] in PLV of different brain region during the successful stop (SS) and failed stop (FS) trials under RHR and LHR inhibitions.

Figure 5A shows the PLVs in F3-F4 signals, which were computed between a minimum of 0.71 and a maximum of 0.83 during successful stop (SS) trails, and between 0.75 and 0.79 in failed stop (FS) trials for the RHR. These findings show increased PLV during failed stop (FS) trials than in successful stop (SS) trails under RHR inhibition. Figure 5B displays the computed PLVs were between 0.72 and 0.79 during successful stop (SS) trails, and between 0.77 and 0.87 during failed stop (FS) for the LHR. These results reveal increased PLV during LHR inhibition in the failed stop (FS) trials than in successful stop (SS) trails in the frontal lobe (F3-F4) of the brain.

Figure 5C displays the higher PLV of EEG signals in LHR (SS) inhibition than in RHR (SS) inhibition in the frontal lobe (F3-F4) of the brain. These PLV findings were investigated in successful stop (SS) trials of RHR and LHR. We found that the increased PLV is associated to human inhibitions of RHR and LHR. The present EEG study clearly shows the response inhibition related PLV features in the frontal lobe (F3-F4) of the brain. Accordingly, eight pairs of EEG channels PLVs were used to classify the neural activities of RHR and LHR inhibitions. The accuracies of the parametric (LDA, QDA) and nonparametric (PARZENDC, KNNC) classification algorithms with optimal PLV features were compared across eight pairs of EEG channels. These eight pairs of EEG electrodes were selected during RHR and LHR inhibitions for the hierarchical classification model, because these electrodes enclosed maximum brain areas included the frontal, central, and temporal and occipital lobes. The frontal, central, and temporal regions have been related to inhibitory control. In this work, we developed a hierarchical classification model of human inhibitions during RHR and LHR.

### 3.3. Architectural Structures of Hierarchical Classification Model

A hierarchical classification model is a system in which lower levels are sorted under a hierarchy of sequentially higher-level units. A basic linear model that did not take into account these sets, it was defective from the onset. However, a hierarchical model allowed us to take into account the effects of these sets, as well as the relations between them [36,37,38]. In the present study, we used the brain connectivity method (i.e., PLV) to hierarchically classify the neural activities of the left and right hand response inhibitions. In the first stage, we classified the EEG signals of successful stop (SS) trials versus failed stop (FS) trials during left-hand response (LHR) inhibition. In the second stage of the hierarchical classification model, we classified the EEG signals of the successful stop (SS) trials versus failed stop (FS) trials under the right hand response (RHR) inhibition. In the third stage, we classified the EEG signals from successful stop (SS) trials in LHR inhibition and successful stop (SS) trials in RHR inhibition. In the final stage, we studied the outcomes of our hierarchical classification model in eight brain cortices related to human inhibition. Figure 6 shows the complete architectural structures of the hierarchical classification model during RHR and LHR inhibitions.

### 3.4. Performance of the Hierarchical Classification Model

This section shows the outcomes of hierarchical classification model in parametric (LDA, QDA) and nonparametric (PARZENDC, KNNC) classification algorithms. The accuracy of the hierarchical model was computed under three conditions: successful stop (SS) vs. failed stop (FS) during RHR inhibition, successful stop (SS) vs. failed stop (FS) during LHR inhibition, and RHR (SS) versus LHR (SS). Figure 6 shows the structure of a hierarchical classification system under the left-hand and right-hand response inhibitions. In this system, the best classification accuracy of 94.44% was obtained in the right temporal-occipital region (T8-O2) of the brain with successful stop (SS) trials during LHR and RHR inhibitions. In addition, Figure 7 clearly presents the performance of hierarchical classification model. Figure 7A demonstrates that the best performance classification accuracy was investigated 88.88% with QDA classifier in the left temporal-occipital lobes (T7-O1) during RHR inhibition. Furthermore, we found other human brain regions also yielded high accuracy with QDA in successful stop (SS) vs. failed stop (FS) trials under RHR inhibition. We investigated 75% accuracy in the frontal cortex (F3-F4), 66.66% accuracy in the frontal-occipital lobes (F4-O1), 61.11% accuracy at the frontal-temporal lobes (F4-T8), 88.88% accuracy in the left tempo-occipital lobes (T7-O1), 61.11% accuracy in the right tempo-occipital lobes (T8-O2), 83.33% accuracy in the temporal lobe (T7-T8), 58.33% accuracy in the left motor cortex (C3-CZ), and 66.66% accuracy in the right motor cortex (C4-CZ). Figure 7B shows the higher accuracy in different brain regions during successful stop (SS) vs. failed stop (FS) under LHR inhibition. We investigated 70.83% accuracy in the frontal cortex (F3-F4), 62.50% accuracy in the frontal-occipital cortex (F4-O1), 61.11% accuracy in the frontal-temporal cortex (F4-T8), 72.22% accuracy in the left tempo-occipital cortex (T7-O1), 88.88% accuracy in the right tempo-occipital cortex (T8-O2), 72.22% accuracy in the temporal cortex (T7-T8), 58.33% accuracy in the left motor cortex (C3-CZ), and 75% accuracy in the right motor cortex (C4-CZ).

Figure 7C shows the eight pairs of electrodes in different areas of the human brain yielded higher accuracy in successful stop (SS) during RHR and LHR inhibitions, i.e., “RHR (SS) vs. LHR (SS)”. We observed 62.5% accuracy in the frontal lobe (F3-F4), 66.66% accuracy in the frontal-occipital lobes (F4-O1), 70.83% accuracy in the frontal-temporal lobes (F4-T8), 83.34% accuracy in the left tempo-occipital lobes (T7-O1), 94.44% accuracy in the right tempo-occipital lobes (T8-O2), 62.50% accuracy in the temporal lobe (T7-T8), 58.33% accuracy in the left motor cortex (C3-CZ), and 75% accuracy in the right motor cortex (C4-CZ). The present results of the hierarchical classification model were averaged values across all the subjects. The best classification performance accuracy was investigated 94.44% with QDA classifier in the right tempo-occipital lobes (T8-O2) of the brain. In this study, the one-way ANOVA test was used to match the classification accuracy between four groups QDA, KNNC, PARZENDC and LDA during LHR and RHR inhibitions. Asterisks show the significant difference in one-way ANOVA between the QDA, KNNC, PARZENDC, and LDA during RHR for SS vs. FS trials [*F* (3, 28) = 4.51, ** *p* < 0.01; Figure 7A].

The one-way ANOVA revealed the significant difference in QDA, KNNC, PARZENDC and LDA under LHR for SS vs. FS trials [*F* (3, 28) = 14.07, *** *p* < 0.001; Figure 7B]. In addition, one-way ANOVA shown the significant difference in QDA, KNNC, PARZENDC and LDA in [*F* (3, 28) = 13.61, *** *p* < 0.001; Figure 7C] during RHR (SS) vs. LHR (SS) trials.

Table 1, Table 2 and Table 3 summarize the sensitivities, specificities, false positive rates (FPR), positive predictive values (PPV) and F-measures only of the best classifiers in terms of accuracy. Table 1 shows the hierarchical classification results during RHR inhibition for SS vs. FS trails. In the left temporal-occipital regions of the brain (T7-O1) the PLVs that were used as optimal features with QDA yielded a higher accuracy of 88.88%, a sensitivity of 0.88, a specificity of 0.88, an FPR of 0.12, a PPV of 0.88, and an F-measure of 0.87, than in LDA, PARZENDC and KNNC. 

In addition, Table 2 displays the hierarchical classification outcomes during LHR inhibition for SS vs. FS trials. In right temporal-occipital regions of the brain (T8-O2) with QDA yielded an accuracy of 88.88%, a sensitivities of 1, a specificities of 0.77, an FPR of 0.23, a PPV of 0.81 and an F-measure of 0.89, than in LDA, PARZENDC and KNNC. 

Moreover, Table 3 presents the hierarchical classification results during RHR (SS) vs. LHR (SS). In right temporal-occipital regions of the brain (T8-O2) with QDA yielded a higher accuracy of 94.44%, a sensitivities of 0.88, a specificities of 1, an FPR of 0, a PPV of 1 and an F-measure of 0.93 than in LDA, PARZENDC and KNNC.

## 4. Discussion

In present study, to the best of our knowledge a hierarchical classification model was developed for the first time to classify the neural activities of human response inhibition, for use as a stop command in BCI technologies as well as in clinical research of ADHD patients. We observed that the QDA had an accuracy of 94.44% in the right temporal-occipital lobes (T8-O2) during LHR and RHR inhibitions. In previous neuroimaging fMRI-EEG study, the right temporoparietal junction (rTPJ) areas were activated during human response inhibition [39]. In addition, former studies have been investigated inhibition related brain regions, such as right inferior frontal gyrus (rIFG) and pre-supplementary motor area (preSMA) [39,40]. In the present study, the PLV method was used to measure the neural coupling between EEG signals from two brain regions because previous study shows the good performance of the classification system when using PLV as a input feature [21,25]. Therefore, in this study, neural connectivity of different brain regions were detected. The optimal phase locking values (PLVs) were used as an input feature in parametric (LDA, QDA) and nonparametric (PARZENDC, KNNC) classification algorithms. Table 4 lists detailed descriptions of the EEG-P300 classification studies found in the latest research literature survey.

An EEG-P300 wave neural marker was observed owing to the reflection of neural activities during LHR and RHR inhibitions. The EEG-P300 wave-related neural activities were identified in the dorsomedial frontal cortex of the brain under human inhibition. However, these P300 waves appeared under response inhibition in all subjects. Changes in the EEG signals of the P300 wave occurred around 300 ms after a stop signal. Earlier studies have reported a significant difference between EEG-P300 waves during successful stop (SS) and failed stop (FS) trials [49,50]. In our study, the positive potential ERP-P300 wave appeared with a frontal-central distribution around 300 ms after the stop stimulus. These brain dynamics results were similar to those in other response inhibition studies [35,51,52]. However, another study also identified P300 wave modulations in a stop-signal task [53]. The ERP-P300 wave in the failed stop (FS) trials were found higher than those in the successful stop (SS) trials under RHR and LHR inhibitions. The higher amplitudes of the ERP-P300 wave were observed in the RHR (SS) than in the LHR (SS) during successful stop trials. Most previous BCI investigations that have been used common method, such as wavelet transform [54], genetic algorithms [55], and common spatial patterns [56] as the optimal feature, whereas in this investigation, PLV was used as an optimal feature because this method has been provided the best overall performance of the system [21]. In our study, a hierarchical classification system was designed for classifying successful stop (SS) and failed stop (FS) trials during RHR and LHR inhibitions.

The leave-one-out method was used to compare the accuracies of parametric and nonparametric classification algorithms [35,57]. With respect to neural connectivity between EEG signals from two electrodes, the PLVs were computed for eight pairs of EEG electrodes in our study, whereas previous studies used only three pairs of EEG channels coupling analysis [21,58,59]. However, the present study has shown the global neural activities of human response inhibition during RHR and LHR. However, earlier studies have been reported short-range and long-range connectivity in different brain regions, which can be used to classify response inhibition features in an EEG-based hybrid BCI system with motor imagery (MI). However, short-range connectivity appeared between electrodes within a particular region, whereas long range connectivity existed between regions that were widely separated. Some authors have argued that whereas long-range connectivity reflects cognitive processing, short-range connectivity may be caused by the volume conduction of neighboring electrodes [25,60]. Therefore, in our study, eight pairs of EEG electrodes, F3-F4, F4-O1, F4-T8, T7-O1, T8-O2, T7-T8, C3-CZ, and C4-CZ, were used. In present work, the effects of small-range connectivity and long-range connectivity were also compared during human response inhibition of RHR and LHR. The best classification performance of PLV feature vectors demonstrated that short-range coupling between right temporal-occipital lobes (T8-O2) provided essential information that was related to response inhibition. The right temporoparietal junction (rTPJ) has been recognized to have an important role in moral judgment and attentional reorienting under response inhibition [39]. Accordingly, we proposed that right temporal-occipital lobes, right inferior frontal gyrus (rIFG) and pre-supplementary motor area (preSMA), can be used as a stop command in a motor imagery-based hybrid BCI model to stop the movement of a wheelchair or robotic arm for a disabled person.

The applications of this study are in development of BCI technologies and in development of clinical research technologies, such as prosthetic hands. In general, motor imagery (MI)-based BCI applications, such as in the movement of a wheelchair a stop command was typically issued through movement of left-hand or right-hand imagination. However, issuing such a stop command, the neural activities (i.e., ERP-P300) of left-hand and right-hand response inhibitions, as a stop command would be more natural than implementing traditional strategies. In rehabilitation studies that were based on a BCI application [61,62,63,64,65,66,67,68] automatic decoders must access not only motor program features, but also signals that represent an intention to move. Therefore, the fact that the extensive literature on BCI has neglected neural associates of response inhibition is remarkable. The ability to detect the neural activities of human inhibition through a hierarchically classification model may represent the first step to stop a BCI system. In addition, the BCI system can utilized the inhibition related neural activity that were classified in our study. The present study has provided a strong sign that closed-loop approaches, in which decoding presentations were feedback, allowed subjects to adapt their inhibition related brain activity to stop Hybrid-BCI system. In addition, inhibition related neural marker (i.e., ERP-P300) can be applied in clinical application to categorize the ADHD patients by inefficient inhibitory control related disorders.

As a final point, it was worth mentioning some limitations of the present study: all subjects were men, which may lead to the problem in comparisons of the result with women subjects. The current experiment design was a well-known stop-signal task, which used only two-dimensional images (2D images) and may not be the most realistic environment for our subjects. A future study may build an even more realistic environment in 3D scenario or virtual reality (VR) to improve the performance and impact of the hierarchical classification model to classify the neural activities of human inhibition under RHR and LHR. The future study will develop a hybrid BCI system, as well as an online system for clinical evaluation of ADHD patients using ERP-P300 neural marker of human inhibition.

## 5. Conclusions

In this study, we have proposed a novel hierarchical classification model to classify the neural activities of the successful stop (SS) and failed stop (FS) trials in RHR and LHR inhibitions. The first time, we developed a hierarchical classification model with ERP-P300 and the PLV coupling method to identify the neural signatures related to human inhibition during RHR and LHR inhibitions. We found that ERP-P300 and PLV could be considered as a reliable EEG feature to classify the neural activities of human response inhibition. In this work, the performance of the basic parametric (LDA, QDA) and nonparametric (PARZENDC, KNNC) classification algorithms were compared to validate the hierarchical classification model during the human inhibitory control of RHR and LHR. The hierarchical classification system achieved the highest accuracy of 94.44% when using the QDA parametric classifier. These EEG-P300 neural activities of RHR and LHR inhibitions could be considered as a stop command in BCI technologies, as well as in the control of the prosthetic hand and in the clinical evaluation of patients with ADHD.

## Figures and Tables

**Figure 1 sensors-19-03791-f001:**
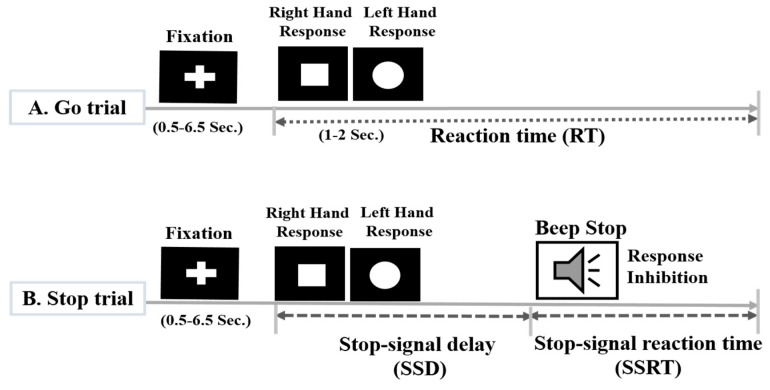
Design of the stop-signal task to investigate the neural activities of human response inhibition. In go trial: a square required a left-hand response (LHR) and a circle required a right-hand response (RHR). In stop trial: a beep sound was used as a stop signal to instruct the participants to withhold their LHR or RHR.

**Figure 2 sensors-19-03791-f002:**
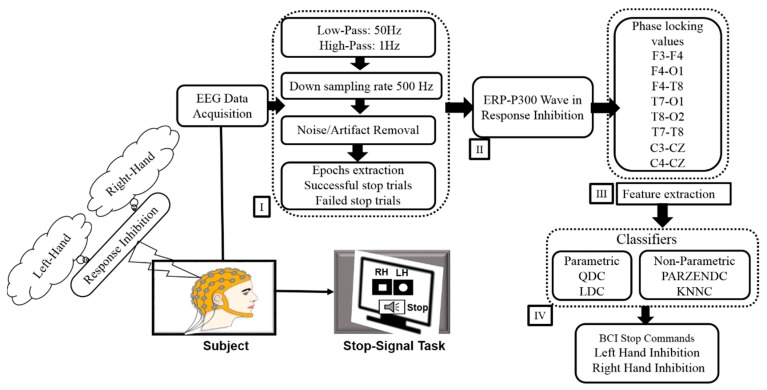
The system architecture of the proposed hierarchical classification model. (I) EEG signal preprocessing steps. (II) Event-related potential (ERP)-P300 wave during LHR and RHR inhibitions. (III) Phase blocking value (PLV) between two EEG signals during LHR and RHR inhibitions. (IV) Parametric (LDA, QDA) and non-parametric (PARZENDC, KNNC) classifiers, which are the most common classifiers used in BCI technologies.

**Figure 3 sensors-19-03791-f003:**
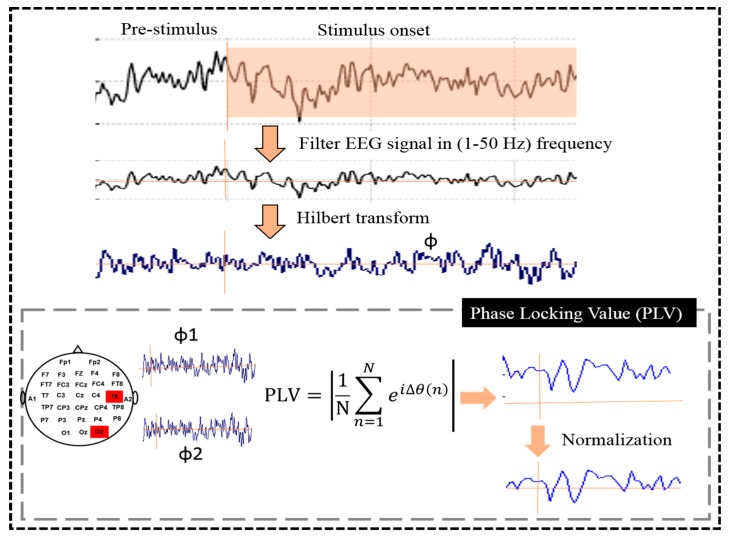
The flowchart of the proposed phase-locking value (PLV) method.

**Figure 4 sensors-19-03791-f004:**
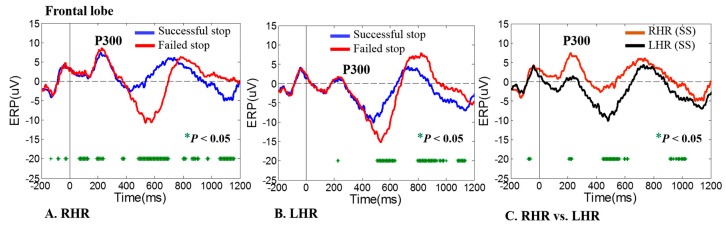
An event-related potential (ERP) of P300 wave was obtained in frontal lobe (FZ) during right-hand response (RHR) and left-hand response (LHR) inhibitions in successful stop (SS) and failed stop (FS) trials. Asterisk display significant difference between SS vs. FS trials by Wilcoxon signed rank test (*p* < 0.05). (**A**) An ERP-P300 wave was investigated in frontal lobe (FZ) during RHR inhibition in SS and FS trials. (**B**) An ERP-P300 wave was observed in frontal lobe (FZ) during LHR inhibition in SS and FS trials. (**C**) An ERP-P300 wave was found during RHR (SS) and LHR (SS). The X-axis displays time in millisecond and Y-axis shows ERP.

**Figure 5 sensors-19-03791-f005:**
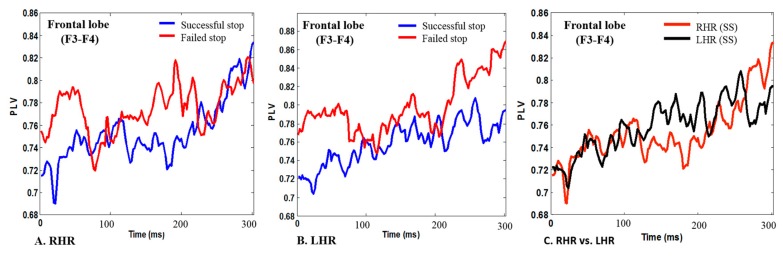
The PLVs were investigated during successful stop (SS) and failed stop (FS) trials in frontal lobe (F3-F4) of the brain. (**A**) The PLV during SS and FS trials in RHR; (**B**) The PLV during SS and FS trials in LHR; (**C**) The PLV during RHR (SS) and LHR (SS).

**Figure 6 sensors-19-03791-f006:**
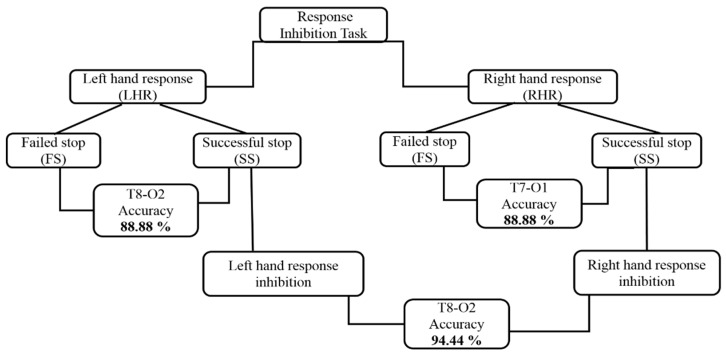
The structure of hierarchical classification system during left-hand and right-hand response inhibitions.

**Figure 7 sensors-19-03791-f007:**
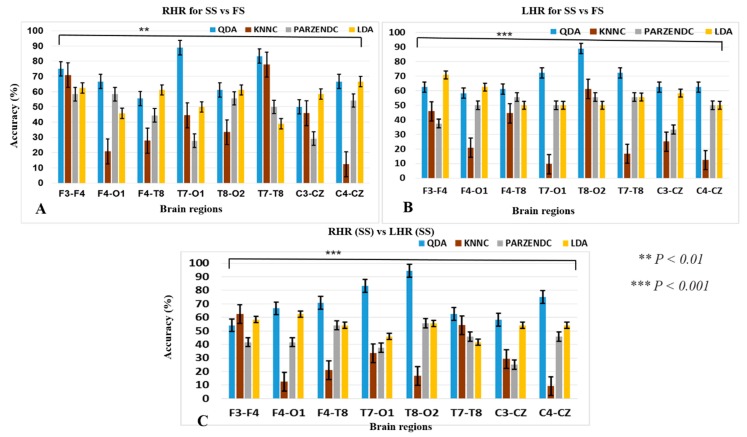
The parametric (LDA, QDA) and nonparametric (PARZENDC, KNNC) classification algorithms performances were compared in accuracy and standard error bars among different brain regions including F3-F4, F4-O1, F4-T8, T7-O1, T8-O2, T7-T8, C3-CZ and C4-CZ. (**A**) Presents the all classifiers outcomes during RHR for successful-stop (SS) vs. failed-stop (FS) trials; (**B**) LHR for SS vs. FS trials; (**C**) RHR (SS) vs. LHR (SS). Asterisks show the significant difference in one-way ANOVA between the QDA, KNNC, PARZENDC and LDA during RHR [*F* (3, 28) = 4.51, ** *p* < 0.01; Figure 7A]. The one-way ANOVA revealed the significant difference in QDA, KNNC, PARZENDC and LDA under LHR [*F* (3, 28) = 14.07, *** *p* < 0.001; Figure 7B], and [*F* (3, 28) s= 13.61, *** *p* < 0.001; Figure 7C] during RHR (SS) vs. LHR (SS).

**Table 1 sensors-19-03791-t001:** Summary of sensitivities, specificities, false positive rate (FPR), positive predictive value (PPV), F-measure and accuracy (%) outcomes during RHR for SS vs. FS.

Outcomes	Hierarchical Model Outcomes Using Brain Connectivity during Right Hand Inhibition							
	F3-F4	F4-O1	F4-T8	T7-O1	T8-O2	T7-T8	C3-CZ	C4-CZ
Classifier	QDA	QDA	LDA	QDA	QDA	QDA	LDA	QDA
Sensitivity	0.58	0.58	0.77	0.88	1.00	0.88	0.83	0.91
Specificity	0.91	0.75	0.44	0.88	0.66	0.77	0.33	0.41
FPR	0.09	0.25	0.56	0.12	0.34	0.23	0.67	0.59
PPV	0.87	0.70	0.58	0.88	0.62	0.80	0.55	0.61
F-measure	0.68	0.63	0.65	0.87	0.76	0.83	0.65	0.74
Accuracy (Mean ± SD)	75 ± 7.46	66.66 ± 11.22	61.11 ± 8.27	88.88 ± 3.32	61.11 ± 7.78	83.33 ± 3.63	58.33 ± 5.69	66.66 ± 5.83

**Table 2 sensors-19-03791-t002:** Summary of sensitivities, specificities, false positive rate (FPR), positive predictive value (PPV), F-measure and accuracy (%) outcomes during LHR for SS vs. FS.

Outcomes	Hierarchical Model Outcomes Using Brain Connectivity during Left Hand Inhibition							
	F3-F4	F4-O1	F4-T8	T7-O1	T8-O2	T7-T8	C3-CZ	C4-CZ
Classifier	LDA	LDA	QDA	QDA	QDA	QDA	QDA	QDA
Sensitivity	0.91	0.66	0.66	0.77	1.00	0.88	0.50	0.83
Specificity	0.50	0.58	0.55	0.66	0.77	0.55	0.75	0.41
FPR	0.50	0.42	0.45	0.34	0.23	0.45	0.25	0.59
PPV	0.64	0.61	0.60	0.70	0.81	0.66	0.66	0.58
F-measure	0.74	0.62	0.62	0.72	0.89	0.75	0.56	0.68
Accuracy (Mean ± SD)	70.8 ± 5.50	62.50 ± 6.02	61.11 ± 6.29	72.22 ± 3.21	88.88 ± 2.48	72.22 ± 2.25	62.50 ± 3.49	62.50 ± 2.65

**Table 3 sensors-19-03791-t003:** Summary of sensitivities, specificities, false positive rate (FPR), positive predictive value (PPV), F-measure and accuracy (%) outcomes during RHR (SS) vs. LHR (SS).

Outcomes	Hierarchical Model Outcomes Using Brain Connectivity in RHR and LHR Inhibitions							
	F3-F4	F4-O1	F4-T8	T7-O1	T8-O2	T7-T8	C3-CZ	C4-CZ
Classifier	KNNC	QDA	QDA	QDA	QDA	QDA	QDA	QDA
Sensitivity	0.58	0.50	0.66	0.91	0.88	0.50	0.66	0.91
Specificity	0.66	0.83	0.75	0.75	1.00	0.75	0.50	0.58
FPR	0.34	0.17	0.25	0.25	0.00	0.25	0.50	0.42
PPV	0.46	0.75	0.72	0.78	1.00	0.66	0.57	0.68
F-measure	0.50	0.29	0.68	0.83	0.93	0.56	0.60	0.77
Accuracy (Mean ± SD)	62.50 ± 2.24	66.66 ± 5.44	70.83 ± 3.52	83.34 ± 1.71	94.44 ± 1.00	62.50 ± 2.70	58.33 ± 1.95	75 ± 2.09

**Table 4 sensors-19-03791-t004:** Descriptions of the EEG-P300 classification method found in the latest literature survey.

EEG Signal	Features	Classifier	References
Sleep stage, Fp1, Fp2	Band power	Hierarchical modal, SVM	[38]
Motor imagery	Band power	adaptive LDA/QDA	[41]
P300	Time points	adaptive LDA/SVM	[42]
P300	Time Points	Co-training LDA	[43]
Motor imagery	Time point, Power spectral	Hierarchical modal, SVM	[44]
Motor imagery	Band power	Hierarchical, KNN, SVM	[45]
P300, Motor imagery	Band power	QDA, LDA, KNNC	[46]
P300	Time points	SWLDA	[47]
Epileptic seizures	Wavelet transformation	Hierarchical modal, SVM	[47]
Virtual-reality (VR)	NWFE, PCA	NBC, KNNC	[48]

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
