# Peer review of "Neural Activities Classification of Human Inhibitory Control Using Hierarchical Model"

_sensors, 2019, doi:10.3390/s19173791_

Round 1
Reviewer 1 Report
This paper is well written and organised. The experimental design and set up are explained throughly and justified with the results. However, it is not clear to see how the Hierarchical Classification Models have been used in Hierarchical way ? Also authors should consider to use other state-of-the-art classification methods and compare the results along with the different feature extraction methods.
Author Response
Dear Reviewer,
Please find the attached file of our responses. Thank you very much for your suggestions.

Reviewer 2 Report
This paper presents an interesting neural activities classification based on Hierarchical model. The experiment is designed properly and the processing method shows good performance and potential. I have some suggestions to improve the manuscript:
(1) When the first time you mentioned F3/FZ/F4/C/T.. electrodes, can be helpful to add a short explanation on the standard EEG electrodes grid.
(2) In Figure 1, the SSD and SSRT should use full word as RT for better understanding.
(3) In 2.3, ICA is used. Do you have any comments on PCA or SVD or the same purpose? Is there a strong reason to choose ICA.
(4) In the results, it seems that the parametric method QDC/LDC is much better than non-parametric method KNNC/PARZENDC. Do you have any comments on the theoretical reason?
(5) Please include benchmarking table to similar work or similar methods.
(6) Part 4. Discussion only has a 4.1, which looks strange...
(7) Some typo needs to be fixed.
Author Response
Dear Reviewer,
Please find the attached file of our responses. Many thanks for your suggestions.

Reviewer 3 Report
This is a potentially interesting study penalized by bad presentation of work.
The text needs considerable english revision and verbs constantly switch from present to past and viceversa, sometimes within the same paragraph.
I am not able to judge the quality of the results because it is not entirely clear to me what the authors did. For example, task description is unclear: are subjects performing motor imagery or actual movements? And, in any case, what kind of movement?
major points:
- lines 61-62: I don't completely agree: motor imagery tasks usually have timings in the same order of magnitude of P300.
- for artifact removal authors mention ICA but it is not clear whether it was performed, and if so which ICA implementation and what criteria were considered to classify bad components.
- Section 2.4 needs improvement. The text is badly written (not english) and unclear. Some parameters are mentioned as if equations were to be introduced, bu they are not, and in the end it is not stated how is PLV calculated.
Moreover, classification algorithms should be detailed with their long name the first time they are mentioned.
- What sw was used to performed the analysis?
- During cross-validation, how many trials were assigned to the training and testing set, respectively?
- Authors refer to their work as "neuroimaging study", however, I am not sure wether surface EEG analysis can be defined as "neuroimaging".
- section 3.3: badly written. Moreover, if figure 6 is reporting average data, as stated, it should also report standar deviation or standard error bars as well as statistics. Same for the tables.
- what are the 2D images mentioned in discussion?
Minor points:
- line 64: command, not commend
- line 101: participating, not participate
- line 120: SSD is mentioned before definition
- line 125: remove "was" and substitute "trials" with "trails"
- remove "the" from sentences straing with "the figure...".
- figure 3 caption: please state clearly what is shown in each panel. Also there are probably two spaces before (ERP).
- lines 254 and 262: incorrect english sentence
- figure 4 captions: panel A and B description are inverted.
Author Response
Dear Reviewer,
Please find the attached file of our responses. Thank you very much for your suggestion.

Round 2
Reviewer 2 Report
Thanks for addressing all my comments. I do not have any further major suggestion. The quality of Fig.2 could be improved.
Author Response
Dear Reviewer,
Please find the attached file of our responses. Thank you very much for your comments on this manuscript.

Reviewer 3 Report
The authors have addressed all the points that I raised. However, the manuscript is still badly written and in some cases the changes made made the text more unclear.
Figures and captions are not very informative.
Statistical analysis should be described in the text and not only mentioned in captions.
Author Response
Dear Reviewer,
Please find the attached file of our responses. We have revised the entire manuscript very carefully according to your comments. Thank you very much for your suggestions and comments on this manuscript.
